# Prevalence and Fracture Risk of Osteoporosis in Patients with Rheumatoid Arthritis: A Multicenter Comparative Study of the FRAX and WHO Criteria

**DOI:** 10.3390/jcm7120507

**Published:** 2018-12-02

**Authors:** Sang Tae Choi, Seong-Ryul Kwon, Ju-Yang Jung, Hyoun-Ah Kim, Sung-Soo Kim, Sang Hyon Kim, Ji-Min Kim, Ji-Ho Park, Chang-Hee Suh

**Affiliations:** 1Division of Rheumatology, Department of Internal Medicine, Chung-Ang University College of Medicine, Seoul 06973, Korea; beconst@cau.ac.kr (S.T.C.); pjh853@hanmail.net (J.-H.P.); 2Division of Rheumatology, Department of Internal Medicine, Inha University College of Medicine, Incheon 22332, Korea; rhksr@inha.ac.kr; 3Department of Rheumatology, Ajou University School of Medicine, Suwon 16499, Korea; serinne20@hanmail.net (J.-Y.J.); nakhada@naver.com (H.-A.K.); 4Division of Rheumatology, Department of Internal Medicine, Ulsan University College of Medicine, Gangneung Asan Hospital, Gangneung 25440, Korea; drkiss@ulsan.ac.kr; 5Division of Rheumatology, Department of Internal Medicine, Keimyung University College of Medicine, Daegu 41931, Korea; mdkim9111@hanmail.net (S.H.K.); okjimin@hanmail.net (J.-M.K.)

**Keywords:** osteoporosis, fracture, fracture risk assessment tool, rheumatoid arthritis

## Abstract

(1) Background: We evaluated the prevalence and fracture risk of osteoporosis in patients with rheumatoid arthritis (RA), and compared the fracture risk assessment tool (FRAX) criteria and bone mineral density (BMD) criteria established by the World Health Organization (WHO). (2) Methods: This retrospective cross-sectional study, which included 479 RA patients in 5 hospitals, was conducted between January 2012 and December 2016. The FRAX criteria for high-risk osteoporotic fractures were calculated including and excluding the BMD values, respectively. The definition of high risk for fracture by FRAX criteria and BMD criteria by WHO was 10-year probability of ≥ 20% for major osteoporotic fracture or ≥ 3% for hip fracture, and T score ≤ −2.5 or Z score ≤ −2.0, respectively. (3) Results: The mean age was 61.7 ± 11.9 years. The study included 426 female patients (88.9%), 353 (82.9%) of whom were postmenopausal. Osteoporotic fractures were detected in 81 (16.9%) patients. The numbers of candidates for pharmacological intervention using the FRAX criteria with and without BMD and the WHO criteria were 226 (47.2%), 292 (61%), and 160 (33.4%), respectively. Only 69.2%–77% of the patients in the high-risk group using the FRAX criteria were receiving osteoporosis treatments. The following were significant using the WHO criteria: female (OR 3.55, 95% CI 1.46–8.63), age (OR 1.1, 95% CI 1.08–1.13), and BMI (OR 0.8, 95% CI 0.75–0.87). Glucocorticoid dose (OR 1.09, 95% CI 1.01–1.17), age (OR 1.09, 95% CI 1.06–1.12), and disease duration (OR 1.01, 95% CI 1–1.01) were independent risk factors for fracture. (4) Conclusions: The proportion of RA patients with a high risk of osteoporotic fractures was 33.4%–61%. Only 69.2%–77% of candidate patients were receiving osteoporotic treatments while applying FRAX criteria. Independent risk factors for osteoporotic fractures in RA patients were age, the dose of glucocorticoid, and disease duration.

## 1. Introduction

Osteoporosis is one of the most well-known complications in patients with rheumatoid arthritis (RA). RA is a disease that presents a state of chronic inflammation that is known to cause an increase in osteoclastic differentiation and an inhibition of osteogenesis [1]. Furthermore, the treatment with glucocorticoids then increases the imbalance which already existed due to the disease. Moreover, this association may be due to a lack of mobility and frequent occurrences of RA during menopause as well as systemic inflammation of RA and the use of corticosteroids [1,2,3]. The prevalence of osteoporosis in patients with RA was reported to be approximately twice as high as in the general population [4]. The frequency of osteoporosis in patients with RA has been reported to be 6.3% to 36.3% in the hip, and 12.3% to 38.9% in the spine [4,5,6]. Compared with controls, the fracture risk in patients with RA also increased for the hip (relative risk (RR): 2) and spine (RR: 2.4) [7]. Moreover, hip and vertebral osteoporotic fractures are known to be associated with an immediate and long-term (up to 20 years) increased risk of mortality [8]. Excess mortality during the first year after a hip fracture ranged from 8.4% to 36%, and the risk of mortality following hip fracture was estimated to be more than 2 times higher than that of the general population [9]. Therefore, it is very important to accurately assess the risk of osteoporotic fractures in RA patients.

The fracture risk assessment tool (FRAX) criteria and the bone mineral density (BMD) criteria established by the World Health Organization (WHO) are widely used for the risk assessment of osteoporotic fractures. The WHO criteria, using BMD measured by dual-energy X-ray absorptiometry (DXA), are the most widely used in the diagnosis of osteoporosis [10]. The management guidelines for the prevention and treatment of osteoporosis, developed by an expert committee of the National Osteoporosis Foundation (NOF), are also based on BMDs [11]. In 2008, a WHO task force introduced the FRAX tool to evaluate the 10-year probability for hip and major osteoporotic fractures [12]. The FRAX model contains various risk factors for osteoporotic fractures including country, age, sex, weight, height, smoking, previous fracture, family history of fracture, glucocorticoid treatment, alcohol intake, and BMD, if available [13]. In particular, RA is the only disease risk factor in the FRAX model, even though the input for RA is just a dichotomous variable.

In clinical settings, physicians determine the proper medications to prevent osteoporotic fractures based on these criteria [11,14]. However, when assessing the risk of osteoporotic fractures in patients with RA, the relevance and benefits among the three assessment methods (WHO osteoporosis criteria and FRAX criteria with BMD and without BMD) have not been clearly studied. Therefore, in this multicenter study, we aimed to evaluate the incidence among a high-risk group for osteoporotic fracture and to identify the risk factors of osteoporotic fractures in patients with RA by comparing these criteria. We also examined the extent of treatments for osteoporosis among patients in need of osteoporosis treatment.

## 2. Experimental Section

### 2.1. Study Population

In this retrospective cross-sectional study, we assessed 479 Korean patients with RA in 5 university hospitals between January 2012 and December 2016. All recruited patients were over the age of 18 and satisfied the 1987 American College of Rheumatology (ACR) criteria or the 2010 ACR/European League Against Rheumatism (EULAR) criteria for RA [15,16]. Recorded data using a medical chart review included age, sex, body mass index (BMI), menopausal status, hormone supplement therapy in postmenopausal women, fracture history, history of parental hip fracture, daily alcohol intake, smoking status, autoantibody status, erythrocyte sediment rate (ESR), C-reactive protein (CRP), and the presence of secondary osteoporosis. The standard of BMI was 25 kg/m^2^ using validated BMI categorization for the Korean population [17]. Therapeutic medication lists for the treatment of RA including current glucocorticoid use, cumulative glucocorticoid dose, and conventional and biological disease-modifying anti-rheumatic drugs, as well as pharmacological intervention for osteoporosis, such as bisphosphonate, selective estrogen receptor modulators (SERM), vitamin D and calcium, were also obtained. The study was approved by the Institutional Review Board (IRB) of each Hospital (C2015163 (1621), 2015-09-026, AJIRB-MED-MDB-15-285, 3-32100191-AB-N-01, and DSMC2015-12-017-007). Informed consent was waived by the IRB.

### 2.2. Evaluation of Osteoporosis by BMD Criteria

Candidates for pharmacological interventions to prevent osteoporosis were assessed using the WHO osteoporosis criteria [10]. All BMD measurements were done using the same technique. The BMD of the lumbar vertebrae (L1–L4) and both hips were measured using DXA (GE Lunar, Madison, WI, USA). *t*- and *z*-scores were calculated with the referent BMD of 5 hospitals. According to the WHO criterion, patients with osteoporosis and osteopenia were defined having a value of the *t*-score that was −2.5 or less, and from −2.5 to −1, respectively, for postmenopausal women or men ≥50 years old. For the evaluation of premenopausal women or men <50 years old, *z*-scores ≤ −2 were considered as osteoporosis.

### 2.3. Osteoporotic Fracture Risk Evaluation by FRAX

The 10-year probability of major osteoporotic and hip fractures was calculated by the FRAX tool, based on medical chart reviews and questionnaires. The FRAX criteria for high risk of osteoporotic fracture were defined as a 10-year probability of ≥20% for major osteoporotic fracture or ≥3% for hip fracture. We calculated two kinds of FRAX values using the Korean model (http://www.shef.ac.uk/FRAX/tool.aspx?country=25); FRAX with BMD was calculated including the femur neck BMD (g/cm^2^) value.

### 2.4. Risk Factors for Fracture

Information regarding previous fractures was obtained through patient self-reporting questionnaires during routine clinic visits and spinal x-ray evaluations. Potential factors associated with osteoporotic fractures were collected by patient interviews, physical examinations, and laboratory tests. Patient baseline characteristics, including clinical variables, RA medications, and osteoporosis-related factors, were investigated at enrollment. The dose of glucocorticoid was calculated on the basis of prednisolone. Glucocorticoid use was grouped as ≥5 mg/day prednisolone equivalent and <5 mg/day prednisolone-equivalent. Candidates for pharmacological interventions preventing osteoporotic fracture were identified using the WHO osteoporosis or the FRAX criteria. We evaluated osteoporotic medications, such as bisphosphonate and SERM, to obtain the ratio of high-risk patients currently undergoing osteoporosis treatments.

### 2.5. Statistical Analysis

All measurements were expressed as means ± SD. Values that were not normally distributed were represented as medians (IQR). A Student’s *t*-test was used for the comparison of mean values. A Pearson’s χ^2^ test was used to compare the differences between categorical variables for analyzing correlations. Multiple linear analysis, including glucocorticoid dose, age, sex, BMI, smoking, and disease duration, was used to identify factors affecting the FRAX values and BMD scores. Glucocorticoid dose, age, BMI and disease duration were continuous variables in this model. A multivariable logistic regression analysis using glucocorticoid dose, age, sex, BMI, and disease duration was performed to obtain the respective odds ratios, and all the variables except sex were continuous variables in this model. In all analyses, a *p* value of <0.05 was considered statistically significant. All statistical analyses were conducted using the SPSS version 20 (IBM Corp, Armonk, NY, USA).

## 3. Results

### 3.1. Baseline Characteristics of the Study Participants

The baseline characteristics of the participants are summarized in Table 1. The data of baseline characteristics in Table 1 were based on the time of BMD test, except for rheumatoid factor and anti-citrullinated protein antibody. The average age of the 479 subjects was 61.5 ± 11.5 years. Study participants included 426 female patients (88.9%) with 353 (82.9%) being postmenopausal. The median disease duration was 53 (33–72) months. The median ESR and CRP levels were 15 (7–29) mm/h and 0.18 (0.05–0.79) mg/L, respectively. The proportion of patients who were using glucocorticoids was 92.3% (442/479), and the median dose was 2.5 mg/day prednisolone-equivalent. The number of patients with biological disease-modifying anti-rheumatic drug (DMARD) use was 56 (11.7%); 13 (2.7%) etanercept, 10 (2.1%) infliximab, 9 (2.9%) adalimumab, 3 (0.6%) golimumab, 11 (2.3%) tocilizumab, 6 (1.3%) abatacept, 2 (0.4%) rituximab, and 2 (0.4%) tofacitinib. A total of 262 (54.7%) patients were receiving osteoporosis treatments: 189 (39.5%) patients were treated with bisphosphonate, and 98 (20.5%) were using SERM. Among all the participants, 81 (16.9%) patients (7 male and 74 postmenopausal females; mean age of 69.5 ± 9.3 years) had fractures, all of which were vertebral.

### 3.2. Distribution of Candidates for Pharmacological Intervention Using the FRAX and the WHO Osteoporosis Criteria

The number of patients with osteoporosis according to the WHO criteria was 160 (33.4%), and the number of candidates for pharmacological intervention using the FRAX criteria with and without BMD was 226 (47.2%) and 292 (61%), respectively. There were significant differences among these three groups (Table 2). When men, women, and postmenopausal women were analyzed separately, the number of candidates for pharmacological intervention was the highest for the FRAX criteria without BMD, followed by the FRAX criteria with BMD, and finally, the WHO osteoporosis criteria (Table 2). The numbers of candidates for pharmacological intervention of women were higher than those of men using the FRAX criteria with BMD (*p* = 0.014) and when using the WHO osteoporosis criteria (*p* = 0.011); however, there were no significant differences when using the FRAX criteria without BMD (*p* = 0.077). The proportion of patients with a BMD < −1.0 was 216/226 (95.6%) among those eligible for FRAX criteria with BMD, and 265/292 (90.8%) for FRAX without BMD.

### 3.3. Proportion of Patients Receiving Osteoporosis Treatment in the High-Risk Group According to the FRAX Criteria and the WHO Osteoporosis Criteria

Table 3 shows the proportion of the patients currently receiving osteoporosis treatments (bisphosphonate or SERM) in the high-risk group, based on the FRAX criteria with or without BMD and the WHO osteoporosis criteria. Of the patients in the high-risk group using the FRAX criteria with BMD, only 77% were receiving osteoporosis treatments, which was similar to the percentages in sex and menopause subgroup analyses (men, 64.7%; women, 78%; postmenopausal women, 78.6%). The proportion of patients undergoing osteoporosis treatment among the high-risk group using the FRAX criteria without BMD was similar (overall, 69.2%; men, 55.6%; woman, 70.6%; postmenopausal women, 71.1%, respectively). However, when applying the WHO osteoporosis criteria, more than 90% of patients were receiving osteoporosis treatments in all patient groups or subgroups (overall, 91.3%; men, 90%; woman, 91.3%; postmenopausal women, 91.3%).

### 3.4. Fracure Risk for the FRAX Criteria, the WHO Osteoporosis Criteria and Fractures

Table 4 shows which dichotomous factors were related to the FRAX criteria with BMD and without BMD, the WHO criteria, and incidence of fractures. Patients with high fracture risk by FRAX with BMD were more likely to be female, especially postmenopausal women (female, *p* = 0.019; menopause, *p* < 0.001), with alcohol use, glucocorticoid use, and proton pump inhibitor use (*p* = 0.042, *p* = 0.007, *p* = 0.018, and *p* < 0.001, respectively), and also with lower BMI (<25 kg/m^2^) (*p* = 0.042). Patients with high fracture risk by FRAX without BMD had lower BMI and longer disease duration, and were more likely to be menopausal women, alcohol users, glucocorticoid users and proton pump users than patients who were not at high risk by FRAX without BMD (*p* = 0.004, *p* = 0.007, *p* < 0.001, *p* = 0.005, *p* < 0.001, and *p* < 0.001, respectively).

The results of frequency comparison for WHO osteoporosis criteria were similar to those for the FRAX criteria with BMD except for glucocorticoid use (female, *p* = 0.017; menopause, *p* < 0.001; BMI < 25 kg/m^2^, *p* < 0.001; alcohol use, *p* = 0.002; proton pump inhibitor use, *p* = 0.008). Biologic use, seropositivity, and ESR elevation did not show any significant difference for all the criteria.

The number of patients meeting the FRAX criteria with BMD was significantly higher in the higher dose with ≥5 mg/day prednisolone group in contrast to the lower dose with < 5mg/day prednisolone group (76/127; 59.8% vs. 136/307; 44.3%, *p* = 0.003).

Multiple linear analyses were implemented to identify factors affecting the FRAX criteria and BMD scores (Table 5). The 10-year probability percentage for hip fractures and that for major osteoporotic fractures in the FRAX with BMD, the 10-year probability percentage for major osteoporotic fractures in the FRAX without BMD, and total hip BMD scores, were significantly associated with glucocorticoid dose, age, female sex, BMI, and disease duration. However, the 10-year probability percentage for hip fractures in the FRAX without BMD was only associated with age and BMI.

Multivariable logistic regression analyses were performed to identify the fracture risk for the FRAX criteria and the WHO osteoporosis criteria, and fractures in patients with RA (Table 6). Glucocorticoid dose (odds ratio (OR) 1.15, 95% confidence interval (CI) 1.03–1.28, *p* = 0.016), age (OR 1.17, CI 1.13–1.2, *p* < 0.001), female sex (OR 4.22, CI 1.8–9.9, *p* = 0.001), and BMI (OR 0.92, CI 0.85–0.99, *p* = 0.002) were independent fracture risk for the FRAX criteria with BMD. Age (OR 1.65, CI 1.47–1.84, *p* < 0.001), female sex (OR 8.23, CI 2.27–29.88, *p* = 0.001), and BMI (OR 0.6, CI 0.5–0.72, *p* < 0.001) were associated with the FRAX criteria without BMD. Likewise, age (OR 1.1, CI 1.08–1.14, *p* < 0.001), female sex (OR 3.55, CI 1.46–8.63, *p* = 0.005), and BMI (OR 0.8, CI 0.75–0.87, *p* < 0.001) were associated fracture risk for the WHO criteria. Furthermore, independent risk factors for fracture are glucocorticoid dose (OR 1.09, 95% CI 1.01–1.17, *p* = 0.02), age (OR 1.09, 95% CI 1.06–1.12, *p* < 0.001) and disease duration (OR 1.01, 95% CI 1–1.01, *p* = 0.002).

## 4. Discussion

In this study, we compared both the WHO and the FRAX criteria to estimate the risk of osteoporotic fractures and to determine candidates for pharmacological treatments. As a result, 33.4% of patients were eligible for the WHO osteoporosis criteria, 47.2% for the FRAX criteria with BMD, and 61% for the FRAX criteria without BMD, with statistically significant differences in each high-risk group for osteoporotic fractures (Table 2). When we performed subgroup analyses separated by men, women, and postmenopausal women, all except premenopausal women showed a similar tendency.

We found that there was a significant difference between the WHO criteria and the FRAX criteria with or without BMD, as well as between the FRAX criteria with BMD and those without BMD, which is different from the previous assertion that there is a high degree of agreement [12,18]. A significant difference in the number of high-risk groups for osteoporotic fractures according to the criteria applied is a serious problem, since both the FRAX and WHO criteria are widely used in clinical practice, and drugs used to prevent osteoporotic fractures have several side effects [11,12,14]. Therefore, it is very important to determine which of the three criteria should be used. To accomplish this, we had to determine the objective standard in order to evaluate all three criteria; however, it was not easy to set standards to which everyone could agree. Ultimately, the goal of osteoporosis treatment is to prevent fractures, so in order to select the best criteria, it may be advantageous to compare the incidences of actual fractures in each criterion. Unfortunately, our study was only a retrospective cross-sectional study, and patients with actual fracture histories were confined to compression fractures of the spine, thereby limiting the prevalence of actual fractures used to evaluate the best criteria.

Recently, a study comparing the FRAX without BMD and BMD alone was published [19]. In that study, the FRAX probability of a major osteoporotic fracture in 50-year-old women was approximately two-fold higher than that for women of the same age, but with an average BMD. The authors concluded that intervention thresholds based on BMD alone do not optimally target women at higher fracture risk than age-matched individuals without clinical risk factors [19]. However, the FRAX with BMD was not evaluated in that study, and there is no comparative study in RA patients. As shown in our study, the WHO criteria may have less actual risk reflected in high-risk disease such as RA [7]. Conversely, the FRAX value may be artificially elevated because RA itself is regarded as one of the major risk factors in FRAX calculations [12].

Table 3 shows that the number of patients actually receiving pharmacological treatments for osteoporosis (bisphosphonate or SERM) was only 69.2%–77% based on the FRAX with or without BMD. Interestingly, more than 90% of patients were receiving medication for osteoporosis when the WHO criteria were applied. This is probably due to the fact that insurance, covering the treatment of osteoporosis in Korea, is based on the WHO criteria.

Osteoporotic fracture, especially hip fracture, can increase mortality rate, so there are attempts to further expand the definition of osteoporosis. For example, while osteoporosis has traditionally been diagnosed based on *t*-score of less than −2.5, osteoporosis can be clinically diagnosed if there is a low-trauma fracture in the absence of other metabolic bone disease, independent of the BMD (*t*-score) value [14]. However, there is no clear evidence that the WHO criteria reflect osteoporotic fractures better than the FRAX criteria. Considering that the FRAX criteria are calculated with BMD values and various clinical aspects of osteoporosis patients, the importance of FRAX criteria in predicting the risk of osteoporotic fracture is increasingly emphasized [19,20], making the decision to treat osteoporosis merely on the basis of WHO osteoporosis criteria problematic. Furthermore, as shown in Table 2, the number of candidates for pharmacological intervention based on the WHO osteoporosis criteria is less than those based on the FRAX criteria; therefore, candidates in need of pharmacological treatment might be missed or their numbers underestimated when applying the WHO criteria. Alternately, we may also consider the new criteria suggested by the National Bone Health Alliance, that osteoporosis may be diagnosed in patients with osteopenia and increased fracture risk using FRAX country-specific thresholds [21]. This is a kind of compromise between FRAX criteria and WHO criteria. In our study, however, almost all patients who met the FRAX criteria had a BMD of −1.0 or less (FRAX criteria with BMD, 95.6%; FRAX criteria without BMD, 90.8%). This result implies that applying the above criteria gives almost the same result as applying the FRAX criteria at least in patients with RA.

Therefore, further study is needed on how to apply the FRAX and WHO criteria in diagnosing and treating osteoporosis, or how to combine these two criteria. In addition, since FRAX criteria with BMD and without BMD may be different, as we have seen in our study, which of these tools is better must be clarified in the future research. Ultimately, osteoporosis treatment should aim for precision medicine. In order to do this, it is necessary to evaluate and approve the risk factors of individual patients. In particular, considering that there is a high risk of osteoporosis in RA patients and that the risk of osteoporotic fracture may be overestimated by FRAX tool in RA patients, more prospective studies involving large populations are required to clarify which assessment tool can precisely estimate osteoporotic fractures in patients with RA.

Nevertheless, it is clear that more than half of postmenopausal women with RA are included in the high-risk group for osteoporotic fractures. The numbers of the high-risk group for osteoporotic fractures in RA patients with postmenopausal woman were higher than those in total RA patients, regardless of which criteria were applied (WHO criteria, 40.3% vs. 33.4%; FRAX criteria with BMD, 55.7% vs. 47.2%; FRAX without BMD, 71.1% vs. 61%, respectively) (Table 2).

The independent risk factors for pharmacological treatments to prevent osteoporotic fractures were female sex, menopause, alcohol use, glucocorticoid use, and proton pump inhibitor use (Table 4). In multiple linear analyses, female sex, older age, lower BMI, longer disease duration, and glucocorticoid dose were associated with a higher probability of osteoporotic fractures (Table 5). In multivariable logistic regression analyses, the OR of female sex was 3.55 and those of age and BMI were 1.1 and 0.8, respectively for the WHO osteoporosis criteria (Table 6). These findings agreed with previous studies [11]. Although smoking has been known to be a major risk factor in many studies [11,22], it was not a statistically significance factor in our study, which was probably due to the fact that the majority of patients enrolled in our study were postmenopausal women with a smoking rate of only 6.9%. Our results showed that the use of biologics, seropositivity, and the levels of acute phase reactants were not associated with an increased risk of osteoporotic fractures. Regarding the use of glucocorticoids, the number of patients meeting the FRAX criteria with BMD was higher in the higher dose with ≥ 5 mg/day prednisolone group. The OR of prednisolone dose was 1.15 in the FRAX criteria with BMD and 1.09 for fractures. These results suggest that no use of glucocorticoid is important in RA patients for the prevention of osteoporotic fractures, and if used, glucocorticoid use should be at a dosage of <5 mg/day.

There are some limitations to this study. The first is that it was conducted not on *all* RA patients, but on RA patients who had a BMD. There may be many patients who had been tested for BMD, given that BMD is not a routine evaluation in RA patients, which could cause a selection bias in this study. The prevalence of the high-risk group of osteoporotic fractures in patients with RA was 33.4%–61.0% in this study, which was higher than the prevalence of osteoporosis in the general population of the United States and Korea [23,24]. However, these results may be overestimated considering selection bias. There are some studies dealing with the incidence and risk factors of osteoporotic fractures in patients with RA [25,26,27]. In a study conducted using the WHO criteria for RA postmenopausal patients, 46.8% of the patients were considered to have osteoporosis (*t*-score ≤ −2.5) [25]. However, in another study using the FRAX criteria for RA patients, in which the population of this study was similar to our study population, only 17.4% of the patients were reported to meet the FRAX criteria for pharmacological interventions [27]. Therefore, a large prospective study of all RA patients is needed to more accurately evaluate the prevalence of the high-risk group of osteoporotic fractures in patients with RA.

Another limitation is that only bisphosphonate and SERM were evaluated as treatments for osteoporosis. Recently, new drugs such as teriparatide and denosumab are being used to treat osteoporosis [11,28,29], but there is no data evaluating these drugs in this study. Nevertheless, this study has important implications, in that bisphosphonate and SERM are still widely used as primary drugs in the treatment of osteoporosis [11,14]. Another limitation is that 88.9% of patients enrolled in this study were female. The final limitation is that our study did not measure vitamin D levels. Lower levels of vitamin D, which are known in addition to regulating bone homeostasis, regulate the differentiation and activities of immune cells [30]. So, vitamin D deficiency can stimulate inflammatory status influencing disease status other than increasing bone erosion. Moreover, mutation and polymorphism in vitamin D metabolism genes are associated with serum 25(OH)D levels [30]. Therefore, these factors might have been associated with the results of our study.

## 5. Conclusions

The proportion of RA patients with a high risk of osteoporotic fractures was 33.4% for WHO criteria, 47.2% for FRAX criteria with BMD, and 61.0% for FRAX without BMD. Only 69.2%–77% of candidate patients were receiving osteoporotic treatments when applying the FRAX criteria, in contrast to 91.3% when applying the WHO criteria. Older age, higher doses of glucocorticoid, and longer disease duration were independent risk factors for osteoporotic fractures in patients with RA.

## Figures and Tables

**Table 1 jcm-07-00507-t001:** Baseline characteristics of the study participants.

Variables	RA Patients (*N* = 479)
Age (year)	61.8 ± 11.5
Sex (female)	426 (88.9%)
Menopause	353 (82.9%)
Weight (kg)	55.4 (49.2–62.8)
Height (cm)	155.3 (150.1–160)
BMI (kg/m^2^)	23.1 (20.9–25)
Smoking	33 (6.9%)
Alcohol	41 (8.6%)
Disease duration (month)	53 (33–72)
1987 ACR criteria	377 (78.7%)
2010 ACR/EULAR criteria	418 (87.2%)
RF titer (IU/mL)	67.2 (24–165.2)
ACPA titer (U/mL)	78 (0.4–179.1)
RF positive	380 (79.3%)
ACPA positive	295 (61.6%)
ESR (mm/h)	15 (7–29)
CRP (mg/L)	0.18 (0.05–0.79)
Medications	
Glucocorticoid	442 (92.3%)
Prednisolone-equivalent dose (mg/day)	2.5 (1.25–5)
NSAID	341 (71.2%)
Biologics	56 (11.7%)
Vitamin D	228 (47.6%)
Calcium	195 (40.7%)
Proton pump inhibitor	123 (25.7%)
Treatment for osteoporosis	262 (54.7%)
Bisphosphonate	189 (39.5%)
SERM	98 (20.5%)
Vertebral fracture	81 (16.9%)
Age (year)	69.5 ± 9.3
Male	7 (8.6%)
Postmenopausal woman	74 (91.4%)

BMI, body mass index; ACR, American college of rheumatology; EULAR, European League Against Rheumatism; RF, rheumatoid factor; ACPA, anti-citrullinated protein antibody; ESR, erythrocyte sedimentation rate; CRP, C-reactive protein; NSAID, non-steroidal anti-inflammatory drug; SERM, selective estrogen receptor modulator. Values were represented as mean ± standard deviation. Values that were not normally distributed are represented as medians (IQR).

**Table 2 jcm-07-00507-t002:** Candidates for pharmacological treatment using the FRAX criteria with and without BMD and the WHO osteoporosis criteria in patients with rheumatoid arthritis.

Patient Groups	FRAX with BMD	FRAX without BMD	Osteoporosis of the WHO	*p* Value ^1^	*p* Value ^2^	*p* Value ^3^
Overall (*n* = 479)	226 (47.2%)	292 (61%)	160 (33.4%)	<0.001	<0.001	<0.001
Men (*n* = 53)	17 (32.1%)	27 (50.9%)	10 (18.9%)	0.006	0.070	<0.001
Women (*n* = 426)	209 (49.1%)	265 (62.2%)	150 (35.2%)	<0.001	<0.001	<0.001
Premenopausal (*n* = 56)	3 (5.4%)	2 (3.6%)	1 (1.8%)	0.659	0.322	0.568
Postmenopausal (*n* = 370)	206 (55.7%)	263 (71.1%)	149 (40.3%)	<0.001	<0.001	<0.001

FRAX, fracture risk assessment tool; BMD, bone mineral density; WHO, World Health Organization. ^1^ FRAX with BMD vs. FRAX without BMD. ^2^ FRAX with BMD vs. osteoporosis. ^3^ FRAX without BMD vs. osteoporosis.

**Table 3 jcm-07-00507-t003:** Current osteoporosis treatments in high-risk groups according to the FRAX criteria and the WHO osteoporosis criteria.

Patient Groups	FRAX with BMD	FRAX without BMD	WHO Osteoporosis
Overall	174/226 (77%)	202/292 (69.2%)	146/160 (91.3%)
Men	11/17 (64.7%)	15/27 (55.6%)	9/10 (90%)
Women	163/209 (78%)	187/265 (70.6%)	137/150 (91.3%)
Premenopausal	0/3 (0%)	0/2 (0%)	1/1 (100%)
Postmenopausal	163/206 (78.6%)	187/263 (71.1%)	136/149 (91.3%)

FRAX, fracture risk assessment tool; BMD, bone mineral density; WHO, World Health Organization.

**Table 4 jcm-07-00507-t004:** Comparison of the frequencies for the high-risk of osteoporotic fractures according to FRAX criteria and WHO osteoporosis criteria.

Variables	FRAX Criteria with BMD (%, *p* Value)	FRAX Criteria without BMD (%, *p* value)	WHO Criteria (%, *p* Value)	Fracture (%, *p* Value)
Sex (female)	48.6%, 0.019	62.2%, 0.077	35.2%, 0.017	17.6%, 0.457
Menopause	57.2%, <0.001	71.1%, <0.001	41.9%, <0.001	21%, <0.001
BMI < 25 kg/m^2^	50%, 0.042	64.9%, 0.004	39%, <0.001	17.3%, 0.868
Disease duration > 2 years	48.9%, 0.098	63.7%, 0.007	35.1%, 0.081	18.6%, 0.063
Seropositivity	46.5%, 0.970	61.4%, 0.170	35.3%, 0.118	17.5%, 0.129
ESR elevation	52.5%, 0.052	65.4%, 0.100	37.4%, 0.196	16.9%, 0.950
Smoking	30.3%, 0.092	45.5%, 0.109	21.2%, 0.209	18.2%, 0.855
Alcohol use	24.4%, 0.007	36.6%, 0.005	9.8%, 0.002	10.3%, 0.239
Glucocorticoid use	49%, 0.018	63.8%, <0.001	33.9%, 0.532	17.3%, 0.81
Biologics use	46%, 0.785	60%, 0.378	40%, 0.137	20.8%, 0.574
PPI use	62.6%, <0.001	77.2%, <0.001	43.1%, 0.008	19%, 0.524

FRAX, fracture risk assessment tool; BMD, bone mineral density; WHO, World Health Organization; BMI, body mass index; ESR, erythrocyte sedimentation rate; PPI, proton pump inhibitor.

**Table 5 jcm-07-00507-t005:** Multiple linear risk analyses using the FRAX criteria and the WHO osteoporosis criteria.

	Variable	β	*p* Value	Adjusted *R*^2^
FRAX with BMD % 10-year probability for hip fracture	Constant	−13.464		0.309
	Glucocorticoid dose	0.418	<0.001	
	Age	0.254	<0.001	
	Female	3.63	<0.001	
	BMI	−0.271	0.001	
	Smoking	2.912	0.012	
	Disease duration	0.01	0.114	
FRAX with BMD % 10-year probability for major osteoporotic fracture	Constant	−24.341		0.407
	Glucocorticoid dose	0.599	<0.001	
	Age	0.386	<0.001	
	Female	7.203	<0.001	
	BMI	−0.163	0.097	
	Smoking	2.289	0.109	
	Disease duration	0.025	0.001	
FRAX without BMD % 10-year probability for hip fracture	Constant	−8.455		0.326
	Glucocorticoid dose	0.179	0.219	
	Age	0.475	<0.001	
	Female	0.515	0.75	
	BMI	−0.641	<0.001	
	Smoking	−0.399	0.832	
	Disease duration	0.004	0.722	
FRAX without BMD % 10-year probability for major osteoporotic fracture	Constant	−26.833		0.630
	Glucocorticoid dose	0.36	<0.001	
	Age	0.547	<0.001	
	Female	7.863	<0.001	
	BMI	−0.411	<0.001	
	Smoking	1.85	0.122	
	Disease duration	0.025	<0.001	
BMD score mean L-spine	Constant	0.705		0.322
	Glucocorticoid dose	−0.037	0.054	
	Age	−0.055	<0.001	
	Female	−0.485	0.024	
	BMI	0.097	<0.001	
	Smoking	−0.102	0.638	
	Disease duration	0.004	0.001	
BMD score total hip	Constant	1.444		0.444
	Glucocorticoid dose	−0.054	0.001	
	Age	−0.047	<0.001	
	Female	−1.072	<0.001	
	BMI	0.111	<0.001	
	Smoking	0.018	0.926	
	Disease duration	−0.004	<0.001	

FRAX, fracture risk assessment tool; BMD, bone mineral density; WHO, World Health Organization; BMI, body mass index.

**Table 6 jcm-07-00507-t006:** Multivariable logistic regression analyses for high-risk of fracture of FRAX criteria and WHO osteoporosis criteria, and fractures in patients with rheumatoid arthritis.

	FRAX with BMD Criteria		FRAX without BMD Criteria		WHO Criteria		Fracture	
	OR (95% CI)	*p* Value	OR (95% CI)	*p* Value	OR (95% CI)	*p* Value	OR (95% CI)	*p* Value
Constant	0		0		0.09		0	
Glucocorticoid dose (mg)	1.15 (1.03, 1.28)	0.016	1.03 (0.94, 1.12)	0.537	1.05 (0.98, 1.14)	0.187	1.09 (1.01, 1.17)	0.02
Age (year)	1.17 (1.13, 1.2)	<0.001	1.65 (1.47, 1.84)	<0.001	1.1 (1.08, 1.13)	<0.001	1.09 (1.06, 1.12)	<0.001
Female	4.22 (1.8, 9.9)	0.001	8.23 (2.27, 29.88)	0.001	3.55 (1.46, 8.63)	0.005	1.81 (0.58, 4.01)	0.395
BMI (kg/m^2^)	0.92 (0.85, 0.99)	0.02	0.6 (0.5, 0.72)	<0.001	0.8 (0.75, 0.87)	<0.001	1.02 (0.95, 1.11)	0.471
Disease duration (month)	1 (1, 1.01)	0.189	1.01 (1, 1.02)	0.121	1 (1, 1.01)	0.093	1.01 (1, 1.01)	0.002

FRAX, fracture risk assessment tool; BMD, bone mineral density; WHO, World Health Organization; OR, Odds Ratio; 95% CI, 95% confidence intervals; BMI, body mass index.

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
