# Peer review of "Prevalence and Fracture Risk of Osteoporosis in Patients with Rheumatoid Arthritis: A Multicenter Comparative Study of the FRAX and WHO Criteria"

_jcm, 2018, doi:10.3390/jcm7120507_

Reviewer 1 Report

The manuscript is clear and well carried out and written.

Despite everything I think that it can be improved :

Although they are not an integral part of the work of analysis carried out of authors, in the introduction some points should be discussed (at least mentioned):

- It is known that RA is a disease that presents a state of chronic inflammation that is known to cause an increase in osteoclastic differentiation and an inhibition of osteogenesis. Furthermore the treatment with glucocorticoids then increases this imbalance already made by the disease state.

- also inadequate levels of vitamin D, which is known in addition to regulating bone homeostasis, also regulates the differentiation and activities of immune cells. vitamin d insufficiency stimulate inflammatory status influencing disease status other then increasing bone erosion (see work: Bellavia D, Costa V, De Luca A, Maglio M, Pagani S, Fini M, Giavaresi G. Vitamin D Level Between Calcium-Phosphorus Homeostasis and Immune System: New Perspective in Osteoporosis Curr Osteoporos Rep. 2016 Oct 13 .) In the analysis, vitamin D level is not taken into account (vitamin D level is among collected data but it is know that is not considered in the FRAX analisis)

 - finally, reference should also be made to possible genetic interferences in the analysis that could distort the results (see Bellavia et al. 2006).

Author Response

Response to Reviewer 1 Comments

NOVEMBER 9, 2018

Manuscript Number: JCM-383919

1. General comments on the revised paper

- We appreciate your critical review of our manuscript. Your comments have helped us to make a better revision. We have revised the paper according to your comments.

2. Replies to reviewer 1

Although they are not an integral part of the work of analysis carried out of authors, in the introduction some points should be discussed (at least mentioned)”

1) It is known that RA is a disease that presents a state of chronic inflammation that is known to cause an increase in osteoclastic differentiation and an inhibition of osteogenesis. Furthermore the treatment with glucocorticoids then increases this imbalance already made by the disease state.

We agree and have added the above to the introduction section (lines 43-45 of introduction).

2) Also inadequate levels of vitamin D, which is known in addition to regulating bone homeostasis, also regulates the differentiation and activities of immune cells. vitamin D insufficiency stimulate inflammatory status influencing disease status other then increasing bone erosion (see work: Bellavia D, Costa V, De Luca A, Maglio M, Pagani S, Fini M, Giavaresi G. Vitamin D Level Between Calcium-Phosphorus Homeostasis and Immune System: New Perspective in Osteoporosis Curr Osteoporos Rep. 2016 Oct 13 .) In the analysis, vitamin D level is not taken into account (vitamin D level is among collected data but it is know that is not considered in the FRAX analysis)

Unfortunately, we had no data about vitamin D level in this study population, so we admit the limitation and added this content in discussion section as follows;

“The other limitation is that our study did not measure vitamin D levels. Lower levels of vitamin D, which is known in addition to regulating bone homeostasis, regulates the differentiation and activities of immune cells. So, vitamin D insufficiency can stimulate inflammatory status influencing disease status other than increasing bone erosion.” (lines 109-113 of discussion).

 3) Finally, reference should also be made to possible genetic interferences in the analysis that could distort the results (see Bellavia et al. 2016).

 We have added following sentence as you recommended.

“Moreover, mutation and polymorphism in vitamin D metabolism genes are associated with serum 25(OH)D levels. Therefore these factors might have been associated with the results of our study.” (lines 113-115 of discussion).

Reviewer 2 Report

Title: Incidence and risk factors of osteoporotic fracture in patients with rheumatoid arthritis: A multicenter comparative study of the FRAX and WHO criteria

 Comments to the Authors:

The authors are submitting a manuscript summarizing a study aimed at assessing the incidence of the high-risk group of osteoporotic fracture and identifying the risk factors of osteoporotic fractures in patients with RA by comparing FRAX and WHO criteria. This study is potentially original, but the authors should address several methodological points.

 Major comments:

1. The study aims to assess the incidence and risk factors of osteoporotic fractures in patients with RA and compare the FRAX criteria and BMD criteria established by the WHO. However, the study is a retrospective cross-sectional study which doesn’t allow assessing the incidence of fractures of being at high fracture risk by FRAX or WHO. Terminology should be corrected. The title of the study also does not accurately reflect the study results.

2. It is not clear how the new fractures were determined. Did you retrieve that information from the chart review? It was indicated that annual self-reported questionnaires and spinal x-rays were evaluated. Was it a standard questionnaire that asks fractures or was any new event remembered by the patient recorded? Also, is annual spinal x-ray a routine practice for all RA patients or was it done only for patients with back pain etc.? If the latter is correct, then asymptomatic spinal fractures might be missed. Moreover, it is interesting that there was no hip fracture among 479 RA patients. It is highly likely that you couldn’t capture those events due to loss to follow-up after a life-threatening event such as hip fracture. Some of the patients with hip fracture might have even died. After all, the new fractures reported in the manuscript seem to underrepresent the actual numbers. 

3. The study includes RA patients who had a BMD at any time between 2012 and 2016. Do you have information about BMD indication? Those patients might have a risk factor for OP other than RA diagnosis. This might be the reason why you did find a higher % of patients with high fracture risk than the Korean general population. This introduces a selection bias. I recommend including patients without BMD.

4. It is not clear when the BMD and FRAX risk calculations were done: before anti-OP treatment? Before a fracture? A random time when all the variables were available? Please clarify this. 

- Also, the study includes patients from 5 different centers. Please clarify if all BMD measurements were done using the same technique.

5. Statistical analysis: The authors conduct multivariable linear and logistic regression analyses to identify risk factors for being at high risk for fractures by FRAX and WHO and also for new fractures. However, all the variables used in the model are already a part of the FRAX risk calculation. Therefore, it is not surprising to find some associations between these variables (e.g., age and sex) and FRAX. I recommend performing multivariable logistic regression analysis for having OP in BMD. Also, other variables which might be associated with OP can be added to the model such as disease activity, seropositivity, acute phase reactants, alcohol, BMI instead of weight, smoking, disease duration, disability, etc.

6. Results: What does table 4 exactly show? The title indicates “Risk factors for high-risk OP fracture…..” However, it is not clear if they were the p values from a multivariable logistic regression analysis or just chi2 test? Also, what was the magnitude of the risk if the results came from a multivariable logistic regression analysis, i.e., OR?

- As indicated above, why did you include only certain variables in the regression analyses? Why didn’t you include BMI instead of weight?

- Please report the descriptive variables as median (IQR) if they were not normally distributed. For example, disease duration, ESR, and CRP, prednisolone dose seem like they were not normally distributed based on the mean and SD.

- If you have data regarding the prior diagnosis of OP, disease activity, disability or any disease severity measure, please report those data as well.

7. The discussion needs improvement. There are some other observational studies examining the role of FRAX and NOF recommendations in determining patients at high risk for fracture and requiring anti-OP treatment. It would be better to include those studies as well. Lastly, the above-mentioned limitations should be addressed by a new analysis, and remaining issues should be added to the limitations.

 Minor comments:

1. Abstract: Please change the term incident as indicated above. Also, please mention that study is a retrospective cross-sectional study. 

2.  Please use 2 decimals for OR throughout the paper.

3. Please change the term high dose and low dose glucocorticoids because technically not all doses ≥5mg/day are considered as high dose glucocorticoids. Moreover, the mean dose of prednisolone in your cohort seems to be 3.3 mg/day.

4. The language of the discussion section is not clear. 

Author Response

Response to Reviewer 2 Comments

NOVEMBER 9, 2018

Manuscript Number: JCM-383919

1. General comments on the revised paper

- We appreciate your critical review of our manuscript. Your comments have helped us to make a better revision. We have revised the paper according to your comments.

 2. Replies to reviewer 2

The authors are submitting a manuscript summarizing a study aimed at assessing the incidence of the high-risk group of osteoporotic fracture and identifying the risk factors of osteoporotic fractures in patients with RA by comparing FRAX and WHO criteria. This study is potentially original, but the authors should address several methodological points.

 Major comments:

1) The study aims to assess the incidence and risk factors of osteoporotic fractures in patients with RA and compare the FRAX criteria and BMD criteria established by the WHO. However, the study is a retrospective cross-sectional study which doesn’t allow assessing the incidence of fractures of being at high fracture risk by FRAX or WHO. Terminology should be corrected. The title of the study also does not accurately reflect the study results.

- We agree and added the term ‘retrospective cross-sectional study’ (line 23 of abstract, line 78 of experimental section, and line 27 of discussion). We also changed the term ‘incidence’ to ‘prevalence’ (title, line 20 of abstract, and line 28 of discussion), as you commented.

 2) It is not clear how the new fractures were determined. Did you retrieve that information from the chart review? It was indicated that annual self-reported questionnaires and spinal x-rays were evaluated. Was it a standard questionnaire that asks fractures or was any new event remembered by the patient recorded? Also, is annual spinal x-ray a routine practice for all RA patients or was it done only for patients with back pain etc.? If the latter is correct, then asymptomatic spinal fractures might be missed. Moreover, it is interesting that there was no hip fracture among 479 RA patients. It is highly likely that you couldn’t capture those events due to loss to follow-up after a life-threatening event such as hip fracture. Some of the patients with hip fracture might have even died. After all, the new fractures reported in the manuscript seem to underrepresent the actual numbers.

- As you pointed out, this study is a retrospective cross-sectional study, so we think it is not appropriate to use the term new fracture. Therefore, the term ‘new fracture’ was replaced with ‘previous fracture’, and the term ‘annual’ was deleted (line 109 of experimental section).

- We also agree that hip fracture is an important part of life-threatening event. In our study, there were no hip fractures, some of the patients with hip fractures might have died or for some other reason we could not find them. So, we described about this point in the discussion section (lines 26-29 of discussion).

 3) The study includes RA patients who had a BMD at any time between 2012 and 2016. Do you have information about BMD indication? Those patients might have a risk factor for OP other than RA diagnosis. This might be the reason why you did find a higher % of patients with high fracture risk than the Korean general population. This introduces a selection bias. I recommend including patients without BMD.

- The data of baseline characteristics in Table 1 were based on the time of BDM test, except rheumatoid factor and anti-citrullinated protein antibody.

- There are indications of BMD testing provided by the National Health Insurance Service in Korea, and there is an item called 'if a patients have a disease is taking medication that can cause osteoporosis, BMD testing can be available'. Therefore, in patients with RA, BMD testing could be carried out at the discretion of the clinician.

- The baseline characteristics of the study population were mean age 61.8 ± 11.5 years, 88.9% of the participants were women, and 82.9% of the women were postmenopausal women. In addition, considering that only 12.9% of patients with premenopausal women or men<50 years old, we think that most of the patients included in this study were patients might have a risk factor such as old ager for osteoporosis rather than RA itself.

- As you pointed out, we think that our study does not represent the entire RA patient, and that there could be a selection bias, and this may be the reason for the overestimated fracture risk in our study. We added this content to the discussion (lines 103-106 of discussion).

 4) It is not clear when the BMD and FRAX risk calculations were done: before anti-OP treatment? Before a fracture? A random time when all the variables were available? Please clarify this. Also, the study includes patients from 5 different centers. Please clarify if all BMD measurements were done using the same technique.

- We added the following sentences;

“The data of baseline characteristics in Table 1 were based on the time of BMD test, except for rheumatoid factor and anti-citrullinated protein antibody.” (lines 131-133 of results).

“All BMD measurements were done using the same technique.” (line 94 of experimental section).

Medication use in Table 1 also indicated the frequency of drug use at the time of the BMD measurement.

 5) Statistical analysis: The authors conduct multivariable linear and logistic regression analyses to identify risk factors for being at high risk for fractures by FRAX and WHO and also for new fractures. However, all the variables used in the model are already a part of the FRAX risk calculation. Therefore, it is not surprising to find some associations between these variables (e.g., age and sex) and FRAX. I recommend performing multivariable logistic regression analysis for having OP in BMD. Also, other variables which might be associated with OP can be added to the model such as disease activity, seropositivity, acute phase reactants, alcohol, BMI instead of weight, smoking, disease duration, disability, etc.

- We performed additional analyzes of seropositivity, disease duration, BMI, and ESR, and added the results to Table 4. Unfortunately, we can’t include disease activity index, such as DAS28, because we had no data about disease activity index in this study population.

- We conducted multivariable linear and logistic regression analyses including other variables which you recommended. We also used BMI instead of weight. BMI and disease duration showed significant values in these analyses, however, seropositivity and the levels of acute phase reactants did not yield significant results. So, the most meaningful model among the various models was selected and the results are presented in Table 5 and Table 6.

- Table 6 contains data on performing multivariable logistic regression analysis for osteoporosis in BMD (WHO criteria). Since the meaning of some associations between variables and FRAX may not be surprising, the discussion emphasized the interpretation of the analysis results for BMD criteria rather than FRAX criteria. (lines 31-32 of abstract, and line 86-87 of discussion).

 6) Results: What does table 4 exactly show? The title indicates “Risk factors for high-risk OP fracture…..” However, it is not clear if they were the p values from a multivariable logistic regression analysis or just chi2 test? Also, what was the magnitude of the risk if the results came from a multivariable logistic regression analysis, i.e., OR? As indicated above, why did you include only certain variables in the regression analyses? Why didn’t you include BMI instead of weight? Please report the descriptive variables as median (IQR) if they were not normally distributed. For example, disease duration, ESR, and CRP, prednisolone dose seem like they were not normally distributed based on the mean and SD. If you have data regarding the prior diagnosis of OP, disease activity, disability or any disease severity measure, please report those data as well.

- Table 4 showed the results of Pearson’s χ2 test (lines 123 of experimental section, and lines 17-18 of results).

- We reanalyzed it using BMI instead of weight, as you suggested. As mentioned above, seropositivity and the levels of acute phase reactants did not show significant results. So the most meaningful model among the various models was selected and the results are presented in Table 5 and Table 6.

- We amended the descriptive variables as median (IQR) if they were not normally distributed in Table 1, as your recommendation.

- Unfortunately, we had no data about disease activity or disability indices in this study population.

 7) The discussion needs improvement. There are some other observational studies examining the role of FRAX and NOF recommendations in determining patients at high risk for fracture and requiring anti-OP treatment. It would be better to include those studies as well. Lastly, the above-mentioned limitations should be addressed by a new analysis, and remaining issues should be added to the limitations.

 - As your recommendation, we reviewed some other articles about the role of FRAX and NOF recommendations in determining patients at high risk for fracture and requiring anti-osteoporotic treatment, and we added these contents to the results and discussion section. (lines 161-162 of results, lines 44-48 of discussion, and lines 56-76 of discussion).

- All revised manuscripts were based on data from new analysis.

 Minor comments:

1) Abstract: Please change the term incident as indicated above. Also, please mention that study is a retrospective cross-sectional study.

 - We mentioned ‘retrospective cross-sectional study’ as your recommendation (line 23 of abstract, line 78 of experimental section, and line 27 of discussion).

 2) Please use 2 decimals for OR throughout the paper.

 - We corrected OR values using 2 decimals.

 3) Please change the term high dose and low dose glucocorticoids because technically not all doses ≥5mg/day are considered as high dose glucocorticoids. Moreover, the mean dose of prednisolone in your cohort seems to be 3.3 mg/day.

 - We changed ‘high-dose’ to ‘higher dose’ and ‘low-dose’ to ‘lower dose’, as your opinion.

 4) The language of the discussion section is not clear.

 - The discussion section was carefully checked again and was clearly modified.

Round  2

Reviewer 2 Report

Title: Incidence and risk factors of osteoporotic fracture in patients with rheumatoid arthritis: A multicenter comparative study of the FRAX and WHO criteria

 Comments to the Authors: Although the authors answered some of my previous comments, there are still issues that are not adequately addressed.

 Major comments:

 1. Abstract: Please briefly define high risk for fracture by FRAX (10-year probability of ≥ 20% for major osteoporotic fracture or ≥ 3% for hip fracture) and BMD criteria by WHO (T score ≤ -2.5 or Z score ≤ -2.0).

 2. The terms “higher and lower dose glucocorticoid” don’t sound appropriate. Please just define as “Glucocortioid use was grouped as ≥ 5 mg/day prednisolone-equivalent and < 5 mg/day prednisolone-equivalent” and replace higher dose with ≥ 5 mg/day prednisolone and lower dose with < 5 mg/day prednisolone everywhere in the manuscript including the tables.

3. Although IQRs are present in Table 1, they should be included as median (IQR) wherever descriptives of those variables are given. Therefore, please provide IQRs of disease duration, ESR and CRP in the results section.

4. The authors indicate that table 4 shows the x2 results of dichotomous variables; however it only shows p values. Moreover, it is not clear what p values compare. For example, BMI<25 for FRAX criteria with BMD; based on the text, I assume p value for that variable indicates patients with high fracture risk by FRAX with BMD had lower BMI than patients who were not at high risk by FRAX with BMD. The table is not giving any information other than p values. What are the percentage of patients who had BMI<25 or ESR elevation etc. in each group? This table needs an edition. Also, while describing the results of table 4, it is not appropriate to name the variables that are significantly different as “risk factors.” It is just a comparison of the frequencies.

For example: “Female sex, especially postmenopausal women, was a risk factor for osteoporotic fracture prediction using the FRAX criteria with BMD (female, p = 0.019; menopause, p < 0.001).” should be described as “Patients with high fracture risk by FRAX with BMD were more likely to be female and postmenopausal, and more frequent alcohol use etc.” Please correct this part accordingly.   

5. Why did you categorize BMI as<25 and="">25? Why didn’t you use WHO categorization? If there is a different validated BMI categorization for the Korean population please describe it in the methods section and a reference for that.

6. Several times, BMD was written as BDM and was not corrected in this revision.

7. Statistical analysis section should be edited and all variables used for linear and logistic regression analysis should be added. Also, please indicate if glucocorticoid dose and BMI used as binary variables or continuous variables in the models. Although, the authors indicate they added new variables to the models, none of them were mentioned in the statistical analysis section.

8. Discusion: I think the following paragraph is irrelevant. Also, considering the suboptimal treatment of OP and fractures in RA patients (based on several other studies), this part is discouraging for OP treatment in RA patients.

“However, serious side effects of anti-osteoporotic drugs should be considered, because the side effects of osteoporosis treatment such as gastrointestinal trouble, osteonecrosis of jaw, and atypical fracture etc. have become a big problem [27]. Furthermore, since RA itself is one of the major items of the FRAX criteria, in patients with RA, the risk of fracture in FRAX criteria may be higher than it actually is.”

9. Discussion still needs extensive editing. I don’t see any significant improvement. I think some of my comments were not very well-understood. For example, the reason why there is a selection bias is not the age and gender of the study population. It is because the authors only evaluated RA patients who had a BMD. BMD is not a routine evaluation in RA patients, correct? Therefore, there must be still lots of patients who did not have BMD. 

Author Response

Although the authors answered some of my previous comments, there are still issues that are not adequately addressed.

 1) Abstract: Please briefly define high risk for fracture by FRAX (10-yearprobability of ≥ 20% for major osteoporotic fracture or ≥ 3% for hip fracture) and BMD criteria by WHO (T score ≤ -2.5 or Z score ≤ -2.0).

- We added the following sentence in the abstract, as per your recommendation (lines 26-28 of the abstract).

“The definition of high risk for fracture by FRAX criteria and BMD criteria by WHO was 10-year probability of ≥ 20% for major osteoporotic fracture or ≥ 3% for hip fracture, and T score ≤ -2.5 or Z score ≤ -2.0, respectively.”

 2) The terms “higher and lower dose glucocorticoid” don’t sound appropriate. Please just define as “Glucocorticoid use was grouped as ≥ 5 mg/day prednisolone equivalent and < 5 mg/day prednisolone-equivalent” and replace higher dose with ≥ 5 mg/day prednisolone and lower dose with < 5 mg/day prednisolone everywhere in the manuscript including the tables.

- The above changes were made. We stated ‘Glucocorticoid use was grouped as ≥ 5 mg/day prednisolone equivalent and < 5 mg/day prednisolone-equivalent’ and replaced ‘higher and lower dose glucocorticoid’ with ‘higher dose with ≥ 5 mg/day prednisolone and lower dose with < 5 mg/day prednisolone’ throughout the manuscript (lines 118-119 of the experimental section, lines 31-32 of the results, and line 82 of the discussion).

 3) Although IQRs are present in Table 1, they should be included as median (IQR) wherever descriptives of those variables are given. Therefore, please provide IQRs of disease duration, ESR and CRP in the results section.

- We added the IQRs of disease duration, ESR and CRP in the results section (lines 141-142 of the results).

 4) The authors indicate that table 4 shows the x2 results of dichotomous variables; however it only shows p values. Moreover, it is not clear what p values compare. For example, BMI<25 for FRAX criteria with BMD; based on the text, I assume p value for that variable indicates patients with high fracture risk by FRAX with BMD had lower BMI than patients who were not at high risk by FRAX with BMD. The table is not giving any information other than p values. What are the percentage of patients who had BMI<25 or ESR elevation etc. in each group? This table needs an edition. Also, while describing the results of table 4, it is not appropriate to name the variables that are significantly different as “risk factors.” It is just a comparison of the frequencies.

For example: “Female sex, especially postmenopausal women, was a risk factor for osteoporotic fracture prediction using the FRAX criteria with BMD (female, p = 0.019; menopause, p < 0.001).” should be described as “Patients with high fracture risk by FRAX with BMD were more likely to be female and postmenopausal, and more frequent alcohol use etc.” Please correct this part accordingly.

- We amended the descriptions of the results of table 4 to make it clear (lines 18-29 of the results).

- We also filled in the percentage of patients corresponding to each variable, and used ‘comparison of the frequencies’ instead of ‘risk factors’ (table 4).

 5) Why did you categorize BMI as<25 and="">25? Why didn’t you use WHO categorization? If there is a different validated BMI categorization for the Korean population please describe it in the methods section and a reference for that.

- Koreans have different physical conditions from Westerners, so the Ministry of Health and Welfare and the National Statistical Office of the Republic of Korea have adopted a standard of obesity for Koreans as BMI 25 kg/m2 (http://kosis.kr/eng/; http://www.index.go.kr/unify/idx-info.do?idxCd=4040). Only 18 patients (3.8%) had a BMI 30 kg/m2 in our study population. Therefore, we applied the Korean obesity criteria of BMI 25kg/m2 in our study, and we added the following sentence in the methods section with a reference; (lines 87-88 of the experimental section)

“The standard of BMI was 25 kg/m2 using validated BMI categorization for the Korean population.”

 6) Several times, BMD was written as BDM and was not corrected in this revision.

 - We corrected BDM to BMD.

 7) Statistical analysis section should be edited and all variables used for linear and logistic regression analysis should be added. Also, please indicate if glucocorticoid dose and BMI used as binary variables or continuous variables in the models. Although, the authors indicate they added new variables to the models, none of them were mentioned in the statistical analysis section.

- As per your recommendation, all variables used for linear and logistic regression analysis were added in the statistical analysis section, and we described that the glucocorticoid dose, BMI and disease duration used in these analyses were continuous variables (lines 127-132 of the experimental section).

 8) Discusion: I think the following paragraph is irrelevant. Also, considering the suboptimal treatment of OP and fractures in RA patients (based on several other studies), this part is discouraging for OP treatment in RA patients.

“However, serious side effects of anti-osteoporotic drugs should be considered, because the side effects of osteoporosis treatment such as gastrointestinal trouble, osteonecrosis of jaw, and atypical fracture etc. have become a big problem [27]. Furthermore, since RA itself is one of the major items of the FRAX criteria, in patients with RA, the risk of fracture in FRAX criteria may be higher than it actually is.”

- We removed the above paragraph.

 9) Discussion still needs extensive editing. I don’t see any significant improvement. I think some of my comments were not very well-understood. For example, the reason why there is a selection bias is not the age and gender of the study population. It is because the authors only evaluated RA patients who had a BMD. BMD is not a routine evaluation in RA patients, correct? Therefore, there must be still lots of patients who did not have BMD.

 - We agreed with your comments, and removed the following paragraphs;

 “In this study, the prevalence of high risk group of osteoporotic fractures in patients with RA was 33.4%–61.0% when using the FRAX criteria with and without BMD and the WHO osteoporosis criteria. This is higher than the prevalence of osteoporosis in the general population of the United States and Korea [17,18]. There are some studies dealing with the incidence and risk factors of osteoporotic fractures in patients with RA [19–21]. In a study conducted using the WHO criteria for RA postmenopausal patients, 46.8% of the patients were considered to have osteoporosis (T-score ≤ −2.5) [19]. In another study using the FRAX criteria for RA patients, only 17.4% of the patients were reported to meet the FRAX criteria for pharmacological interventions [21].”

 “Another limitation is that 88.9% of patients enrolled in this study were female. Considering the gender distribution in patients with RA [31], this indicates that this study does not represent the entire RA patient population. This could introduce a selection bias and might be the reason of overestimation of the fracture risk in our study than those in Korean general population. However, considering that the risk of osteoporosis and osteoporotic fractures are higher in women, (especially postmenopausal women), and that this study was based on actual practice data obtained from multi-centers, our study may well represent the practical problems in a real world.”

 - We added the following paragraph; (lines 86-99 of the discussion)

 “The first is that this study was conducted not on all RA patients, but on RA patients who had a BMD. There may be many patients who did not have BMD tested, given BMD is not a routine evaluation in RA patients, which could cause a selection bias in this study. The prevalence of high risk group of osteoporotic fractures in patients with RA was 33.4%–61.0% in this study, which was higher than the prevalence of osteoporosis in the general population of the United States and Korea [24,25]. However, these results may be overestimated considering selection bias. There are some studies dealing with the incidence and risk factors of osteoporotic fractures in patients with RA [26–28]. In a study conducted using the WHO criteria for RA postmenopausal patients, 46.8% of the patients were considered to have osteoporosis (T-score ≤ −2.5) [26]. However, in another study using the FRAX criteria for RA patients, in which the population of this study was similar to our study population, only 17.4% of the patients were reported to meet the FRAX criteria for pharmacological interventions [28]. Therefore, a large prospective study of all RA patients is needed to evaluate more accurately the prevalence of high risk group of osteoporotic fractures in patients with RA.”

 Thank you for considering our manuscript for publication in Journal of Clinical Medicine.

We look forward to hearing from you.

Yours sincerely,